# Biomarkers of Fabry Nephropathy: Review and Future Perspective

**DOI:** 10.3390/genes11091091

**Published:** 2020-09-18

**Authors:** Tina Levstek, Bojan Vujkovac, Katarina Trebusak Podkrajsek

**Affiliations:** 1Institute of Biochemistry, Faculty of Medicine, University of Ljubljana, Vrazov trg 2, 1000 Ljubljana, Slovenia; tina.levstek@mf.uni-lj.si; 2Centre for Fabry Disease, General Hospital Slovenj Gradec, Gosposvetska cesta 1, 2380 Slovenj Gradec, Slovenia; bojan.vujkovac@sb-sg.si; 3Clinical Institute for Special Laboratory Diagnostics, University Children’s Hospital, University Medical Centre Ljubljana, Vrazov trg 1, 1000 Ljubljana, Slovenia

**Keywords:** biomarkers, Fabry nephropathy, Fabry disease, nephropathy, genomics, epigenomics, metabolomics, proteomics, transcriptomics

## Abstract

Progressive nephropathy is one of the main features of Fabry disease, which largely contributes to the overall morbidity and mortality burden of the disease. Due to the lack of specific biomarkers, the heterogeneity of the disease, and unspecific symptoms, diagnosis is often delayed. Clinical presentation in individual patients varies widely, even in patients from the same family carrying the same pathogenic *GLA* variant. Therefore, it is reasonable to anticipate that additional genomic, transcriptomic, proteomic, and metabolomics factors influence the manifestation and progression of the disease. The aim of this article is to provide an overview of nephropathy in Fabry patients and the biomarkers currently used in the diagnosis and follow-up. Current biomarkers are associated with late signs of kidney damage. Therefore, there is a need to identify biomarkers associated with early stages of kidney damage that would enable early diagnosis, which is crucial for effective treatment and prevention of severe irreversible complications. Recent advances in sequencing and -omics technologies have led to several studies investigating new biomarkers. We will provide an overview of the novel biomarkers, critically evaluate their clinical utility, and propose future perspectives, which we believe might be in their integration.

## 1. Introduction

Fabry disease (FD; OMIM#301500) is an X-linked lysosomal storage disorder associated with inherited or *de novo* disease causing variants in the α-galactosidase A gene (*GLA*; OMIM*300644) [1,2]. Reduced or even absent α-galactosidase A (α-Gal A; EC 3.2.1.22) activity leads to accumulation of glycosphingolipids with terminal α-D-galactosyl residues, especially globotriaosylceramide (Gb3) and globotriaosylsphingosine (lyso-Gb3) in plasma, urine and different organ systems [1,3,4], mainly cardiac, renal, endothelial and neuronal [5]. The major physiological source of Gb3 is globoside, a glycolipid of erythrocytes and cells membranes found in different tissues [6]. FD is rare disorder with the estimated incidence in the general population between 1 in 40,000 and 1 in 117,000 [7]. However, based on recent newborn screening studies, the prevalence in some populations was reported to be markedly higher with 1 in 1300 to 1 in 7800 males [8].

Pathophysiological processes start at a very young age [9], but most patients do not show symptoms in infancy. Gb3 deposition has been detected in fetuses [10] and in newborns with classical FD phenotype [9]. Men with classical phenotype, who have no or very low α-Gal A activity (<1% of mean normal), often develop first symptoms in early childhood, including acroparesthesias, angiokeratomas, intolerance to heat, hypo/anhidrosis, cornea verticillate and gastrointestinal disturbances [6,11,12]. The disease then gradually progresses to major organ failure in early adulthood, such as progressive kidney disease, cardiac signs, and symptoms and cerebrovascular events [6,11,12]. On the other hand, men who have a significant residual α-Gal A activity typically present with only one organ involvement [11,12]. Although, FD is an X-linked trait, women can also develop typical symptoms due to random X-inactivation, which result in a mosaicism of gene expression, leading to differential expression of the functional or mutant enzyme [13]. Inactivation of the mutant allele leads to a milder phenotype, while the inactivation of the wild-type allele leads to more severe phenotype with an earlier onset [14]. The unaffected cells secrete mostly 46 kDa mature form of the α-Gal A and not the high-uptake mannose 6-phosphorilated form. The mature form is able to complement the activity in the population of cells lacking the expression of the enzyme [15]. Clinical presentation in women therefore varies from an asymptomatic or mild, later onset phenotype to a phenotype similar to that of men with a classical phenotype [16].

The diagnosis of FD in men is based on reduced α-Gal A activity in plasma, leukocytes, or dried blood spots [17,18,19], but in women, this measurement is unreliable, because enzyme activity in female with FD may not be elevated [20]. Gb3, the substrate of α-Gal A and the main accumulation deposit in FD, was considered as a possible diagnostic marker of FD. Gb3 accumulates mainly within cells, while it circulates in the extracellular space within lipoproteins [21]. However, there is no strict correlation between plasma Gb3 concentrations and clinical manifestation in Fabry patients [22,23]. Therefore, Gb3 level in urine and plasma are only suitable for identifying males with classical phenotype. Elevated plasma lyso-Gb3, the deacylated derivative of Gb3, has been designated as a hallmark of FD [3] and allows better differentiation between patients with classical and non-classical phenotype and healthy subjects [4,24,25]. Therefore, this biomarker may improve the diagnosis of clinically relevant FD, particularly in females with normal or borderline α-Gal A activity [24]. However, false negative results were also reported in non-classical phenotypes [3,24]. After the introduction of enzyme replacement therapy (ERT), patients with classical phenotypes show a rapid decrease in lyso-Gb3 concentration, whereas the decrease is slower in men with non-classical phenotype and women [26,27]. The serum lyso-Gb3 may serve as a marker for tissue involvement to assess which heterozygotes should be considered for treatment despite normal α-Gal A activity [24]. The gold standard for the diagnosis of female Fabry patients is still *GLA* gene sequencing, which is also important in men for the confirmation of the pathogenic *GLA* variant and prediction of the disease phenotype [28]. Early diagnosis of FD is important to initiate Fabry disease specific treatment, namely with ERT [27] or chaperones [29].

## 2. Current Diagnostic Approaches and Follow-Up of Fabry Nephropathy

Before the era of kidney transplantation and dialysis, end-stage renal disease (ESRD) was the leading cause of premature death in patients with FD [7]. However, progressive nephropathy remains one of the main manifestations of FD, which usually leads to ESRD in untreated patients with classical phenotype from the third to the fifth decade of life [30,31]. The rate of progression of chronic kidney disease (CKD) is similar to that of diabetic nephropathy [32]. The Fabry Outcome Survey reported a baseline prevalence of nephropathy in 59% of men and 38% of women with FD [7]. Even though progression to ESRD is less common in women with FD and are less likely to progress from moderate CKD to ESRD, the median age at which patients reached ESRD was 38 years regardless of their gender [16,30]. Renal involvement contributes largely to the overall burden of morbidity and mortality in FD [11].

Gb3 accumulation occurs in various types of renal cells, including podocytes, mesangial, and interstitial cells, cells of the proximal and distal tubules, and loop of Henle, as well as vascular endothelial cells and smooth muscle cells [33]. The understanding of the pathophysiological mechanisms leading to CKD and ESRD is still scarce and further research is needed to clarify the complexity of genotype-phenotype correlation.

The main approaches used in nephrology to diagnose patients with FD are kidney biopsy, high-risk population testing, and family screening [34]. FD should be considered in patients with CKD without a clear cause of nephropathy [8]. For diagnosis, assessment of renal involvement, and monitoring of treatment, biomarkers of renal damage are often evaluated (albuminuria/proteinuria, serum creatinine, glomerular filtration rate (GFR), and cystatin C) together with urinary microscopy and renal biopsy.

### 2.1. Albuminuria/Proteinuria

Albuminuria/proteinuria is clinically most often used biomarker of Fabry nephropathy, although kidney damage is present already in the non-albuminuric state. Therefore they are not sensitive biomarkers for early kidney damage, as biopsies of normoalbuminuric Fabry patients have already shown advanced lesions [35]. However, proteinuria is one of the most important markers for monitoring the progression of nephropathy in treated and untreated patients, since patients with higher proteinuria show a much steeper decline in renal function over time [30,32]. Proteinuria has been shown to stimulate interstitial inflammation and fibrosis. Furthermore it promotes tubular cells to undergo partial epithelial mesenchymal trans-differentiation, which induces cell-cycle arrest and promotes the release of fibrogenic cytokines [36,37].

### 2.2. Serum Creatinine and Glomerular Filtration Rate (GFR)

Regular assessment of renal function in Fabry patients includes use of measured and estimated glomerular filtration rate (eGFR). Due to inaccuracy of creatinine-based GFR measurements, it is recommended to use measured GFR measurements (e.g., iohexol GFR) at least annually [28]. Because these methods are more complex and tedious, estimated GFR measurements using appropriate formula are more widely used. Currently used serum creatinine-based equations in the clinical practice are the Chronic Kidney Disease Epidemiology Collaboration (CKD-EPI) equation for adults [38] and the Schwartz formula for children [39]. Some Fabry patients develop GFR loss before development of proteinuria [16]. Glomerular hyperfiltration may be a common feature in young Fabry patients and may be considered as an early marker of Fabry nephropathy [40], which masks impairment of renal function, although rare, a decline in GFR can already be seen in adolescence [41]. Slope of progression of renal insufficiency was correlated to the level of proteinuria and in addition, it was not linear as shown by Schiffmann et al., where they retrospectively analyzed the decline of eGFR. The slope of renal function with an eGFR of more than 60 mL/min/1.73 m^2^ was −3.0 in men and −0.9 mL/min/1.73 m^2^/year in women, and with an eGFR of less than 60 mL/min/1.73 m^2^ it was −6.8 and −2.1 mL/min/1.73 m^2^/year in men and women, respectively [42].

### 2.3. Cystatin-C

Cystatin-C, a cysteine protease inhibitor, is constantly produced by all nucleated cells. It is freely filtered by the glomerulus and then reabsorbed and catabolized in the tubular epithelial cells. Therefore, it does not re-enter the bloodstream or urine. Cystatin-C concentration was found as a superior and more sensitive marker than serum creatinine for detecting early renal dysfunction and small decreases in GFR in Fabry patients of both genders. Therefore, it could be valuable as a prognostic marker and for estimating the efficiency of the ERT [43]. Probably the main reason why cystatin-C is not widely used is that it is more costly, time-consuming, and less available than creatinine.

### 2.4. Urine Microscopy

As a non-invasive, reasonably priced, and expeditious diagnostic tool, urine microscopy can be useful for the diagnosis and assessment of FD progression. Most of the cells present in the urine of Fabry patients are renal tubular epithelial cells [44]. Mulberry cells with characteristic “Maltese cross bodies” (oval fat bodies) can be detected in the urinary sediments of Fabry patients under a polarized microscope [45,46]. Example of a Maltese cross in the urine sediment of FD patients is shown in Figure 1. Furthermore, there is a specific morphological population of Maltese cross bodies, characterized by a lamellarized appearance with protrusions, probably due to Gb3 resembling “mosquito coils”, which most likely represents a fragmentation of shed nephronal epithelial cells with accumulated lysosomal Gb3 [47]. In addition their excretion correlated with the concentration of albumin in urine and could therefore be useful for accessing Fabry nephropathy burden [47]. Despite the fact, that this method could represent a valuable tool it is not commonly used, because it requires special equipment (phase contrast microscope) and well-trained personnel.

### 2.5. Renal Biopsy

Kidney biopsy with electron microscope analysis is recommended for all individuals with CKD, a *GLA* variant of unknown significance, and an uncertain diagnosis of FD to rule out other comorbidities, as it is currently the only reliable diagnostic method for confirming or excluding Fabry nephropathy [48]. Significant histologic changes occur before the typical clinical signs of CKD therefore, findings in biopsy are crucial for choosing the optimal therapeutic strategy and follow-up in high-risk patients [49,50,51,52]. Example of a kidney biopsy under electron microscopy with typical lamellar inclusions in a 27 year old Fabry male patient with normal kidney function (eGFR 102 mL/min/1.73 m^2^), normoalbuminuria (albumin to creatinine ratio (ACR) 12 mg/g), but high levels of podocyturia (urinary podocytes (uPod) 2420/g creatinine) is shown in Figure 2. Electron microscopic studies demonstrate typical osmophilic bodies, called myeloid or Zebra bodies, packed with lamellated membrane structures [33]. In contrast to electron microscopy, diagnosis could be much more challenging with light microscopy techniques, since the majority of stainings cause a washout of lipid contents. As a result, more unspecific vacuolization of podocytes and epithelial cells is a characteristic histological finding [33,53]. Mesangial expansion, segmental and global glomerulosclerosis, tubular atrophy, and intestinal fibrosis are also present at an early stage [33,53,54]. To better classify the extent of tissue involvement, the International Study Group of Fabry Nephropathy developed a scoring system of histological changes on light microscopy and toluidine blue–stained semithin sections [50]. The score quantifies the Gb3 density deposition in glomeruli, interstitium, and vessels, as well as progressive lesions (glomerulosclerosis, ischemic glomeruli, and tubulointerstitial fibrosis). Recently, bedside stereomicroscopy for Fabry kidney biopsies has been recommended as a complementary method to current histologic evaluation as it was demonstrated to have high diagnostic sensitivity for FD [55].

## 3. Novel Biomarkers for Predicting Development and Progression of Fabry Nephropathy

Heterogeneous phenotype, varying disease severity and symptoms onset make the diagnosis of FD notoriously difficult and often delayed for several years. Early diagnosis is crucial, as treatment should begin before the kidneys are irreversibly damaged. ERT was reported to reduce Gb3 deposits from most renal cells, but to a lesser extent in podocytes [56], and to stop or slow down the progression of Fabry nephropathy [57,58,59]. However, once proteinuria occurs, it usually does not normalize despite treatment [60]. In addition, in patients with overt proteinuria or reduced GFR (<60 mL/min/1.73 m^2^), ERT does not prevent further deterioration of renal function [8]. Therefore, there is a considerable need for novel biomarkers that would enable identification and also prediction of development and progression of Fabry nephropathy.

Patients with rapid progression of kidney disease have higher urinary protein to creatinine ratio [61] and additionally, underlying conditions, such as hypertension, hyperlipidemia, and smoking, may contribute to a progressive loss of GFR [62]. However, environmental factors and glycolipid accumulation due to the disease causing *GLA* variants cannot fully explain the phenotypic variability. Even between the same family members carrying the same pathogenic *GLA* variant, there is enormous clinical variability [63]. It is therefore reasonable to anticipate that additional unknown biochemical, genetic, and epigenetic factors (modifiers) may influence the rate of progression of Fabry nephropathy. Recently, new biomarkers (bikunin, tubular proteins) have been proposed to improve the assessment of renal impairment, but further research is needed to evaluate their clinical utility. Various -omics approaches have been used to identify disease-associated biomarkers that could be used for diagnosis, prediction of disease progression and monitoring of treatment. Despite ongoing efforts to identify FD biomarkers, there is still no proper plasma or urine biomarker.

### 3.1. Bikunin

Bikunin or urinary trypsin inhibitor is a serine protease inhibitor present in plasma and many tissues. It is excreted in urine and its levels were found significantly higher in Fabry patients with nephropathy; therefore, it may serve as a biomarker of renal impairment in FD [64]. The origin of the higher bikunin levels may imply direct renal involvement and secondary activation in response to the storage of glycosphingolipids in biochemical pathways associated with inflammation; however, further studies are needed to elucidate the mechanisms involved in the elevation of urinary bikunin levels [64]. Yet, renal impairment alone is not sufficient to explain higher urinary bikunin levels, as no correlation between serum creatinine and urinary bikunin levels was found [64].

### 3.2. Tubular and Glomerular Proteins

Impaired glomerular and tubular function in Fabry patients is reflected in an abnormal urinary excretion of tubular and glomerular proteins. Therefore, these biomarkers could be a valuable tool to assess kidney involvement and predict the progression of Fabry nephropathy. Larger studies are needed to thoroughly investigate the sensitivity and their correlations with Fabry nephropathy progression, the biomarkers currently used, and the changes in response to ERT.

Aguiar et al. found that the biomarkers of glomerular (transferrin and type IV collagen) and tubular (α1-microglobulin, N-acetyl-β-glucosaminidase, and alanine aminopeptidase) dysfunction were elevated even in a subgroup of patients without clinical signs of kidney disease. Furthermore, more significant correlation with eGFR was reported for type IV collagen and N-acetyl-β-glucosaminidase as it was for albuminuria [65]. Besides increased N-acetly-β-D-glucosaminidase, Shiffmann et al. also reported an increase in β2-microglobulin [66]. A decrease in the glomerular marker IgG, the tubular markers α1-microglobulin and retinol-binding protein as well as the shared tubular and glomerular markers albumin and transferrin in a population of 13 women with FD after long-term ERT was reported [67]. Furthermore another study showed normalization of urinary excretion of uromodulin after ERT and reduction in untreated Fabry patients [68].

### 3.3. Urine Podocytes

The earliest sign of renal damage appears to be podocyte foot process effacement, whereas it was found in young classic FD patients without clinically evident signs of Fabry nephropathy [35,69,70]. Podocytes as terminally differentiated cells do not divide; therefore, their replacement potential in adult is limited. Podocytes accumulate Gb3 more than other renal cell types [33]. Accumulation in podocytes continues until the third decade [71]. Alongside their volume continues to increase, resulting in an increase in podocyte foot process width and podocyte loss [71]. Injured podocytes detach from the glomerular basement membrane and are lost to urine. Podocyturia therefore correlates with the clinical severity of Fabry nephropathy [70]. A recent study of podocyte glycocalyx damage showed that podocalyxin loss may be associated with reduced adhesion of podocytes to the extracellular matrix, which enables detachment and urinary excretion [72]. Podocyturia disrupts glomerular permselectivity, causes albuminuria/proteinuria and leads to glomerulosclerosis and fibrosis [73,74]. A positive correlation was found between podocyturia and ACR [75]. Despite podocyturia is an early clinical sign of kidney injury and could serve as a diagnostic test to assess kidney involvement, it is still not regularly used in clinical practice. This is mainly because methods for assessing podocyturia are not yet standardized and not available in the majority of clinical settings.

### 3.4. Promising Aproaches in Discovery of Fabry Nephropaty Biomarkers

#### 3.4.1. Genomics

To date, there are no reported genome-wide association studies (GWAS) that would evaluate genetic modifiers associated to the specific clinical presentation of FD. However, studies reported the association of genetic variants in inflammatory and coagulation factor genes with an increased risk of cerebral lesions and stroke in Fabry patients [76,77,78]. Genetic variants in human alcohol dehydrogenase family genes, namely, *ADH4* (rs1126670, rs1126671, rs2032349) and *ADH5* (rs2602836) correlated with FD progression, despite ERT [79]. However, we are not familiar with any studies investigating association of specific genetic variants and Fabry nephropathy. Genetic modifiers are likely to contribute to organ involvement and disease progression as clinical presentation itself varies widely between family members with identical pathogenic *GLA* variant. Therefore, the identification of genetic modifiers is a promising tool for a more personalized prediction of kidney involvement and nephropathy progression. This could make it much easier for clinicians to decide when to start treatment.

#### 3.4.2. Transcriptomics

There are some limited studies focused on the selected urinary miRNA expression in FD. Decrease of certain miRNA species, namely, miR-29 and miR-200, have been associated to the renal fibrosis prior to onset of pathological albuminuria [80,81]. In serum, specific miRNAs species, namely, miR-1307-5p, miR-21-5p, miR-152-5p, and miR-26a-5p, were significantly down-regulated in Fabry patients with ERT, compared to those without. Additionally, miR-19a-3p and miR-486-5p were reported to be down-regulated only in male patients with ERT [82]. The analysis of miRNA expression is not commonly used in the clinical practice, due to the lack of standardized protocols. However, we expect miRNA analysis to become an important tool in routine applications, as the accessibility, high specificity, and sensitivity of miRNAs are promising for the identification of useful biomarkers [83].

#### 3.4.3. Epigenomics

The exact influence of DNA and/or histone modifications on the disease course, especially on nephropathy, remains unclear. To date, research on epigenetics is very limited and there are few studies that investigate epigenetic changes in FD [84,85]; however, to the best of our knowledge non in Fabry nephropathy. In Fabry patients, high concentrations of the methylated/non-methylated Gb3 isoform were found in urine compared to controls characterized by the absence or trace amounts of the methylated Gb3 isoform in control urine samples [86]. Epigenetic changes could be reversible, which means they could provide an opportunity for therapeutic development.

#### 3.4.4. Proteomics

Proteomics has also been applied to identify new potential biomarkers. The study of the urinary proteome is the most widely used approach, as urine collection is simple, non-invasive, and unrestricted, allowing large quantities to be collected [66]. Moreover, biomarkers of kidney damage, which reflect abnormal nephron function, are expected to be found mainly in urine. Urinary proteome analysis is less difficult, since urine contains less proteins than serum or plasma. Proteome studies are summarized in Table 1. Several altered proteins were identified (either up- or down-regulated), but their correlation with disease course and organ involvement remains to be elucidated. To the best of our knowledge, there are no reported studies specifically aiming to study urinary proteome of Fabry patients with kidney involvement in order to identify specific biomarkers for Fabry nephropathy. It can be assumed that new proteome biomarkers will complement the albumin/protein level in urine. Proteome analysis is more complex than genome analysis and therefore has the potential to elucidate complex diseases such as FD. Nowadays, large-scale, high-throughput analyses are available that can also detect less abundant proteins [87]. Even in proteomics, there is a lack of standardized protocols, which hampers its clinical utility. In addition, data processing and analysis require a high level of expertise.

#### 3.4.5. Metabolomics

Metabolites represent a promising biomarker as their levels change rapidly in response to physiological changes [66]. Moreover, the metabolome comes closest to the functional phenotype and changes in the metabolite level were observed, even if no changes in the level of proteins and transcripts were detectable. To reduce interindividual variation, the experimental design must be carefully considered [99]. Metabolomics studies, summarized in Table 2, identified several altered analogs/isoforms, in urine and plasma, structurally related to Gb3 and lyso-Gb3 [100,101,102]. To our knowledge, however, no one has yet investigated the changes in the concentration of metabolites specifically in Fabry nephropathy. Analogs refer to the sphingosine moiety modifications, while isoforms contain various fatty acid [86]. Lyso-Gb3 is a promising biomarker as it is directly involved in the pathogenesis of the disease through the stimulation of inflammation and fibrosis by inducing the release of secondary mediators of glomerular damage in diabetic nephropathy [103,104]. In addition, lyso-Gb3 promotes the proliferation of vascular smooth muscle cells [3] and may be a potential driver of CD80 expression [21], suggesting that these cells act as antigen presentation cells and are involved in inflammatory pathways [105]. Additionally, CD80 interaction with integrins has been associated with podocyte migration, detachment, and podocyturia [21]. In cultured podocytes, Lyso-Gb3 induced the expression of tumor growth factor β1 (TGF-β1) by Notch1 activation and the expression of CD74 by activation of the macrophage migration inhibitory factor [103,106,107]. TGF-β1 promotes fibrosis in response to chronic inflammation by promoting the synthesis of extracellular matrix in renal cells via epithelial-to-mesenchymal transition [106]. Activation of CD74 further promoted the release of inflammatory cytokines [107]. Unfortunately, no obvious correlation was found between lyso-Gb3 concentrations and renal failure or elevated urinary albumin or protein levels [4,108]. Interestingly, the relative elevation of lyso-Gb3 and its analogs in urine and plasma differ significantly. Lyso-Gb3, but not the analogs, are more elevated in plasma. On the contrary, in urine lyso-Gb3 analogs are generally more elevated [109]. Recently, total urine concentration of lyso-Gb3 and its analogs were found specific to FD [110]. Urinary galabiosylceramide (Ga2) analogs/isoforms were found to be more increased in Fabry patients compared to healthy controls [111]. Results show that in pediatric patients, gender strongly influences the urinary level of metabolites, with men having higher level of all metabolites [112].

## 4. Discussion and Future Perspective

The introduction of ERT in 2001 has dramatically improved the management of FD; however, FD is still an untreatable disease that has a major impact on patients’ quality of life. Moreover, Fabry nephropathy is an important cause of morbidity and mortality in men with classical phenotype and also in some women. The clinical presentation in individual patients is highly variable, even in patients from the same family carrying the same pathogenic *GLA* variant. Therefore, further studies are needed to elucidate additional factors that influence the manifestation and progression of the disease. The identification of these factors and related pathways is important for the identification and possible subsequent prevention of the metabolic changes that could lead to disease progression and the development of chronic complications.

Considering recent advances in sequencing approaches, it is reasonable to anticipate, that novel genomic and transcriptomics markers influencing the development and progression of the nephropathy are going to be identified. Of course, studies focusing on identification of genetic modifiers will need to include a large cohort for reliable conclusions, which is an important drawback when studying rare diseases, such as Fabry disease. Due to the rarity of the disease, the number of samples is often limited, which hampers the statistical analysis. Additionally, long-term follow-up samples might significantly improve the yield of the transcriptomics studies. They could provide further insights into the pathophysiology of Fabry nephropathy and could help to open an additional path in developing novel therapeutic strategies. Previous studies have also shown the importance of including adult and pediatric patients of both genders [93,112,113], which further underlines the need for a large cohort. Therefore, multicenter collaborations are strongly recommended while they would provide a larger number of samples and thus improve statistical evaluation.

The metabolic origin and rate of progression of Fabry nephropathy resembles that of diabetic nephropathy. Since several studies have already been conducted in the field of CKD and diabetic nephropathy [119], translating of the knowledge of biomarkers suspected of being involved in the development and progression of kidney failure could contribute to a better understanding of the molecular mechanisms of Fabry nephropathy.

Markers of glomerular and tubular damage are promising to detect renal damage in the normoalbuminuric stage, while currently used biomarkers, namely, albuminuria and low GFR, are markers of late renal injury. Similarly, given that podocyte foot process effacement seems to be one of the earliest signs of renal damage in FD [69], biomarkers reflecting podocyte injury merit further investigation. The development of new -omics technologies has led to new opportunities in the search for novel biomarkers. Despite several promising biomarkers, so far none of them have been translated into clinical practice. All these potential biomarkers require large, longitudinal observational studies to validate the candidate biomarkers before they are implemented into routine clinical practice. Furthermore, it is highly unlikely that the biomarkers discovered with -omics technologies will alone be sufficient to reliably predict the development and/or progression of Fabry nephropathy. Multifactorial models that would integrate clinical, genetic, and biochemical factors [120] are more likely to provide the evidence needed to translate this knowledge into clinical practice. Currently, there are no models reported that would aim to predict the Fabry disease development and progression. Even in CKD, where the number of the patients worldwide is enormously higher, currently available studies that are combining different factors influencing its progression are based merely on blood biochemical values to predict the risk of progression to dialysis [121]. The models combining genetic and biochemical markers, along with clinical characteristics, in order to predict the development and progression of renal disease have not been reported.

Nevertheless, we believe that an attempt to establish a multivariable prediction model to achieve greater sensitivity and specificity—which would integrate clinical, genomic, epigenetic, and biochemical data—might be promising in achieving the final goal, namely, the identification of higher risk patients in their preclinical stages of chronic complications of FD and the development of specific prevention strategies. It could also provide a potential pathway for the development of treatment, thereby reducing the morbidity and mortality of FD.

## Figures and Tables

**Figure 1 genes-11-01091-f001:**
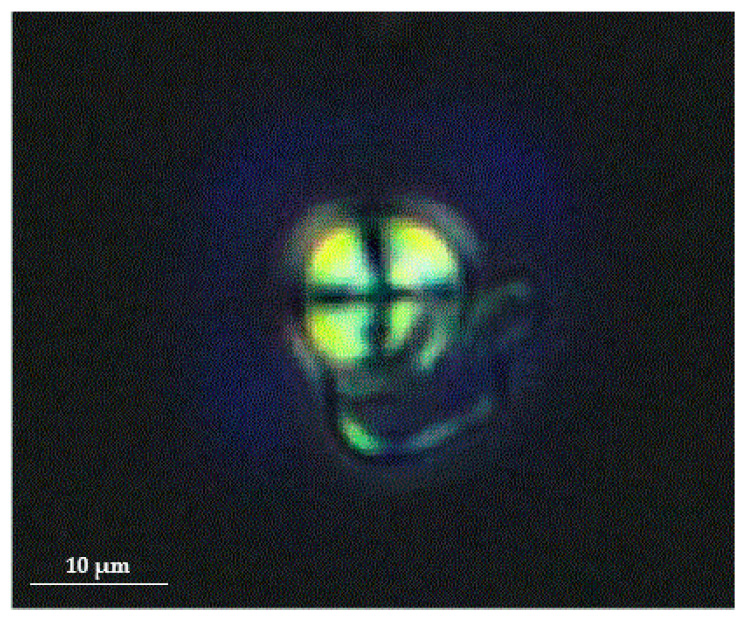
Birefringent Maltese cross in the urine sediment of Fabry patient when viewed under a polarized microscope (magnification 400×). Figure courtesy of Mravljak M; Department of Internal Medicine, General Hospital Slovenj Gradec.

**Figure 2 genes-11-01091-f002:**
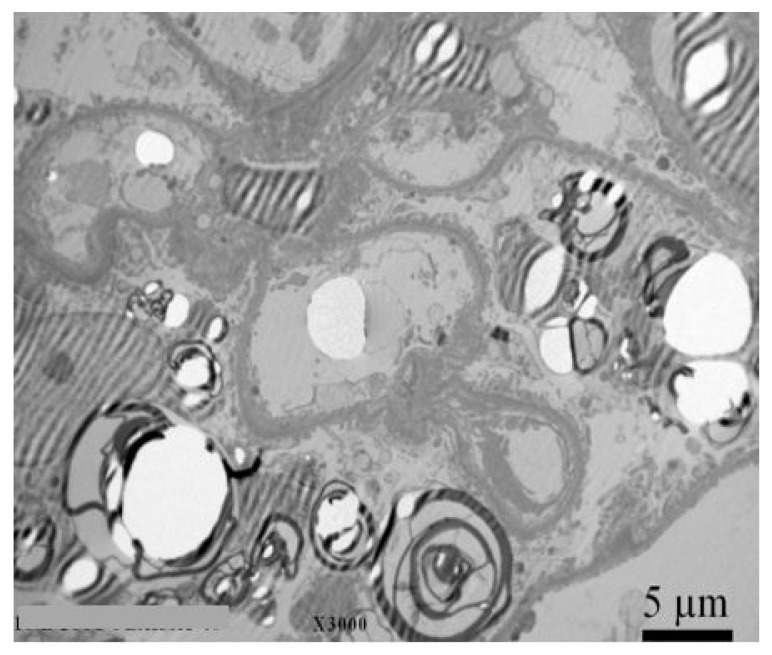
Kidney biopsy with electron microscopy with typical lamelar inclusions in a 27-year old Fabry male patient with normal kidney function (eGFR 102 mL/min/1.73 m^2^), normoalbuminuria (ACR 12 mg/g), but high levels of podocyturia (UPod 2.420/g creatinine). Diffuse and numerous myeloid inclusions in podocyte cytoplasm (black arrow) with presence of vacuoles in the cytoplasm of podocytes (white arrow) are evident. Figure courtesy of Pleško J and Kojc N; Institute of Pathology, Ljubljana Medical Faculty.

**Table 1 genes-11-01091-t001:** Proteomic studies in FD.

Subjects	Sample Type	Methods Used	Outcomes in FD	Reference
69 FD patients and 83 healthy subjects	plasma	electrochemi-luminescence assays	FD patients: elevation of FGF2, VEGFA, VEGFC and IL-7. Non-classical FD patients: significant correlation between IL-7 and residual enzyme activity.	[88]
66 FD patients and 10 healthy subjects	urine	LC-MS/MS	Early stage and asymptomatic FD patients: elevation of albumin, uromodulin, α1-antitrypsin, glycogen phosphorylase brain form, endothelial protein receptor 1 and intracellular adhesion molecule 1.	[89]
7 FD patients and 7 healthy controls	urine-derived cells	Label-free quantitative nLC-MS/MS	46 candidate proteins with altered expression indicating an involvement of the Coordinated Lysosomal Expression and Regulation (CLEAR).	[90]
8 classic FD patients	plasma	2D electro-phoresis and MALDI-TOF-MS and MS/MS	Short-term ERT: reduction of 15 plasma proteins involved in inflammation, oxidative and ischemic injury, or complement activation. Pre-ERT: significant elevation of plasma β-actin, inactivated complement C3b (iC3b), and C4B. Longer-term ERT: gradually decreased iC3b.	[91]
11 naïve Fabry patients, 12 permanently ERT-treated patients and 12 healthy subjects	second morning urine	free approach coupled with LC-MS/MS analysis	Up-regulation: uromodulin, prostaglandin H2 D-isomerase and prosaposin.	[92]
32 FD patients and 14 healthy subjects	plasma	MS iTRAQ followed by MRM approach	Male FD patients: specific and sensitive panel of eight proteins (22 kDa protein, afamin, 𝛼1 antichymotrypsin, apolipoprotein E, 𝛽-Ala His dipeptidase, hemoglobin 𝛼-2, isoform 1 of sex hormone-binding globulin, and peroxiredoxin 2). Female FD patients: a nine-protein marker panel; where three proteins (apolipoprotein E, hemoglobin 𝛼-2, and peroxiredoxin 2) were common to both genders.	[93]
8 FD and 6 healthy subjects	PBMC	2D electro-phoresis and MALDI-TOF-MS	Downregulated: calnexin, rho GDP dissociation inhibitor 2, rho GDP-dissociation inhibitor 1, chloride intracellular channel protein 1. Up-regulated: g-enolase, 14-3-3 protein theta, 14-3-3 protein zeta/delta, and galectin-1.	[94]
10 male pediatric FD patients	urine	QTOF-MS	Prosaposin: significantly elevated in FD in and reduced after 12 months of ERT. GM2AP: elevated in the pretreatment Fabry patients and reduced after 12 months of ERT.	[95]
35 treatment-naïve female FD patients, 89 healthy subjects	urine	CE-MS	Urinary biomarker profile for female patients: diagnostics of the FD, monitoring of ERT.	[96]
20 FD patients, 10 healthy controls	urine	SDS-PAGE and MALDI-TOF MS	Differentially expressed proteins: α-1-antitrypsin, α-1-microglobulin, prostaglandin H2 D-isomerase, complement-c1q tumor necrosis factor-related protein, and Ig kappa chain V–III.	[97]
13 pediatric FD patients	serum	LC-MS/MS	Reduced after 6 months of ERT: α2-HS glycoprotein, vitamin D-binding protein, transferrin, Ig- α-2 C chain, and α-2-antiplasmin.	[98]

nLC-MS/MS, nano liquid chromatography couple to tandem mass spectrometry; MALDI-TOF-MS, matrix-assisted laser desorption/ionisation-time of flight tandem mass spectrometry; MS/MS, tandem mass spectrometry; MS-iTRAQ, mass spectrometry isobaric tags for relative and absolute quantification; MRM, multiple reaction monitoring, QTOF-MS, quadrupole time-of flight-coupled to mass spectrometry; CE-MS, capillary electrophoresis coupled to mass spectrometry; SDS-PAGE, sodium dodecyl sulphate-polyacrylamide gel electrophoresis.

**Table 2 genes-11-01091-t002:** Metabolomic studies in FD.

Subjects	Sample Type	Methods Used	Outcomes	Reference
42 FD patients, 48 healthy controls	random urine	LC-MS/MS	The total urinary concentration of lyso-Gb3 and its analogues: 100% specific for classical and non-classical FD patients.	[110]
18 asymptomatic females, 18 symptomatic females, 27 males, 16 control urines, 58 control plasmas	plasma and urine	UPLC-MS/MS	Urinary ceramide dihexoside (CDH): more prominent glycosphingolipid in females elevated isoforms of Ga2 with 10 different longer chain can detect asymptomatic heterozygotes better than Gb3 and lyso-Gb3.	[113]
34 untreated and 33 treated Fabry males, 54 untreated and 19 treated females, 34 males and 25 female healthy control	random urine	UPLC-MS/MS	Validated method for separation of 12 most abundant Ga2 isoforms/analogs from their lactosylceramide (LacCer) counterparts: urinary LacCer significantly higher in female than males after normalization with creatinine, higher the Ga2(C24:0) urinary level in the untreated males comparing to the the untreated female	[114]
15 untreated and 28 treated Fabry males, 21 untreated and 10 treated Fabry females, 15 males and 26 female healthy control	plasma	UPLC-MS/MS	Validated UPLC-MS/MS method for the multiplex analysis of lyso-Gb3 and its 6 analogs.	[115]
16 untreated Fabry males, 16 healthy Fabry males	random urine	TOF MS	Untreated FD patients: 22 urinary Ga2 isoforms/analogs; quantification of Ga2 and Gb3 urinary isoforms/analogs that were elevated in untreated Fabry males.	[111]
9 males with classic FD, 7 males with later-onset FD, 10 females, 5 males with functional variants, 40 healthy controls	plasma	nano-LC-MS/MS	Plasma lyso-Gb3: higher in all subgroups of FD, especially in patients whose disease stage had proceeded. Detection of eight lyso-Gb3-related analogues: lyso-Gb3(-12) and lyso-Gb3(+14) for the first time. Majority of the FD patients had elevated plasma concentrations of the lyso-Gb3 analogues, especially lyso-Gb3(-2).	[116]
55 pediatric Fabry patients, 26 healthy pediatric controls, 108 adult Fabry patients, 16 healthy adult controls	random urine	UPLC-MS/MS	Pediatric FD patients: lyso-Gb3 (+16) the most reliable for the diagnosis of pediatric FD females, whereas lyso-Gb3 (−12), lyso-Gb3(+16) and lyso-Gb3 (+34) for males children had lower urinary lyso-Gb3 levels than adults.	[112]
74 Fabry patients, 41 healthy controls	plasma	UPLC-MS/MS	Lyso-Gb3 and related analogues in plasma: higher in FD males compared to females higher in untreated males compared to treated males decreased after the beginning of ERT in a Fabry male.	[109]
114 Fabry patients, 34 healthy controls	plasma	TOF MS	Fabry patients: three new lyso-Gb3 analogs (m/z ratios at 802, 804, and 820) with higher relative concentration in males compared to female patients. None was detected in the majority in healthy controls.	[102]
24 Fabry patients, 8 healthy controls	plasma	TOF MS	Identification and characterization of five novel analogues/isoforms of Gb3	[101]
164 Fabry patients, 94 healthy controls	random urine	HPLC-MS/MS	Further analysis of 7 previously discovered lyso-Gb3-related biomarkers: not detected in healthy controls higher excretion levels in Fabry males compared to females correlation with ERT status in males	[117]
16 male Fabry patients, 16 healthy control males	random urine	TOF MS	15 isoforms/analogs of Gb3: variable urinary quantities of Gb3-related metabolites in untreated Fabry males identification of methylated Gb3-related analogs	[118]
63 untreated Fabry patients, 59 healthy controls	random urine	TOF MS	Seven novel urinary biomarkers (mass-to-charge ratios (m/z) of 758, 774, 784, 800, 802, 820, and 836) none were detected in controls higher concentrations in males with FD compared to females. decreased excretion of all lyso-Gb3-related analogs after ERT in male FD patients	[100]

MS, mass spectrometry; TOF, time-of-flight; LC-MS/MS, liquid chromatography coupled to tandem mass spectrometry UPLC-MS/MS, ultra-performance liquid chromatography coupled to tandem mass spectrometry; HPLC-MS/MS, high-performance liquid chromatography coupled to tandem mass spectrometry.

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
