# Peer review of "Biomarkers of Fabry Nephropathy: Review and Future Perspective"

_genes, 2020, doi:10.3390/genes11091091_

Round 1
Reviewer 1 Report
This is a comprehensive review of the renal involvement if Fabry disease (FD) with a specific focus on biomarkers. It is a well-written review but there are a few issues to address as I outlined below:
1) Please add a scale bar to Figure 1.
2) Authors need to be more specific on histological diagnosis in Figure 2. They could use arrows to indicate the depositions or changes. Or, an EM image belonging to a healthy control would be helpful. Authors also cold think about adding a toluene blue-stained section since it is a validated method for FD diagnosis on renal histology.
3) Sections 2 and 3 can be combined since Section 2 is relatively short and its content (i.e., Fabry nephropathy) is also discussed in Section 3. Authors can keep the title of Section 3 and move the paragraphs belonging to Section 2 under Section 3 (those can be the starting sentences of Sections 3).
4) Section 4 needs to be restructured: Subsections 4.1 and 4.2 are all about the biomarkers whereas the contents of 4.3 to 4.8 are not related to the section title (those are mainly signatures of disease or techniques/approaches to find out reliable biomarkers rather than novel biomarkers).
5) Please correct the following typo errors:
- 1 in 40.000 to 1 in 117.000 >> 1 in 40,000 to 1 in 117,000 (pg. 1)
- 1 in 1.300 to 1 in 7.800 >> 1 in 1300 to 1 in 7800 (pg. 1)
- acroparasthesias >> acroparesthesias (pg. 2)
- m2 >> make ‘2’ superscript (pg. 4)
- lamellated >> lamellated (pg. 4)
- technics >> techniques (pg. 4)
- Galabiosylceramid >> Galabiosylceramide (pg. 8)
- rela-tive >> relative (pg. 10)
- globotriaosylceramid >> globotriaosylceramide (Abbreviations, pg. 17)
Minor points:
1) Please do the following corrections in the related sentences as suggested:
- propose future perspective >> propose future perspectives (Abstract)
- tubular cells of the proximal and distal tubules >> cells of the proximal and distal tubules (pg. 2)
- for confirming or rejecting >> for confirming or excluding (pg. 4)
- kidney biopsy with electron microscopy >> kidney biopsy under electron microscopy (pg. 4)
- irreversible damaged >> irreversibly damaged (pg. 5)
- GLA disease causing variants >> disease causing GLA variants (pg. 5)
- detach in urine >> detach from the glomerular basement membrane (GBM) and are lost to urine (pg. 6)
- proteome of Fabry patients >> urinary proteome of Fabry patients (pg. 7)
- physiologic changes >> physiological changes (pg. 7)
- in vitro cultured podocytes >> cultured podocytes (pg. 8)
- large enough cohort >> large cohort (pg. 8)
- studying rare disease >> studying rare diseases (pg. 8)
- Studies on proteomics >> Proteomic studies (title of Table 1, pg. 9)
- d-isomerase >> D-isomerase (Table 1, pg. 9)
2) The following sentences are very long and confusing, please re-write (dividing each into two sentences would be great):
- In contrast to electron microscopy … (pg. 4)
- While the clinical presentation … (pg. 8)
3) The following sentence does not sound right, please re-write:
- Although urine contains … (pg. 7)
4) Please define the following abbreviations when those appear first in the text: GLA (pg. 1), FD (pg. 1), ACR (pg. 4), UPod (pg. 4), creat (pg. 4) and GWAS (pg. 6).
5) Authors can save space (and make the other columns wider) if they get rid of the author names and keep only the reference numbers in the first columns of Tables 1 and 2.
Author Response
Please see the attachment. Thank you and best regards, Katarina Trebusak podkrajsek

Reviewer 2 Report
The article is correctly written. Would require a slight shortening.
Some of the statements seem unnecessary (e.g. the information about Dr. factories and Anderson is common knowledge).
- In the Introduction section, I propose to enter information about the specificity of FD in women
Beck M, Cox TM. Mol Genet Metab Rep. 2019 Oct 22;21:100529.
- for practical purposes, it is worth adding information about the usefulness of lyso Gb3 in medical practice (prognosis, evaluation of treatment results.
Nowak A, Mechtler TP, Desnick RJ, Kasper DC. Mol Genet Metab. 2017 Jan-Feb;120(1-2):57-61
Author Response
Please see the attachment. Thank you and best regards, Katarina Trebusak Podkrajsek
